# Synergistic antifungal activity of antiretrovirals with amphotericin B against *Aspergillus species*

**Ammar A. Khan**[1,2], **Ehab A. Salama**[1,2], **Mohamed N. Seleem**[1,2]*

**1** Department of Biomedical Sciences and Pathobiology, Virginia-Maryland College of Veterinary Medicine, Virginia Polytechnic Institute and State University, Blacksburg, Virginia, United States of America, **2** Center for One Health Research, Virginia Polytechnic Institute and State University, Blacksburg, Virginia, United States of America

* seleem@vt.edu

## Abstract

Aspergillosis is a life-threatening fungal infection that primarily affects the lungs of immunocompromised individuals, including those living with human immuno-deficiency virus (HIV), and is associated with mortality rates exceeding 50%. The infection is predominantly caused by *Aspergillus fumigatus*, a pathogen that has become increasingly difficult to treat due to the emergence of azole-resistant strains. Although azoles have traditionally served as first-line antifungals, the rise in resistance has necessitated broader use of amphotericin B (AmB), a polyene agent whose clinical utility is limited by its considerable toxicity. To address this therapeutic challenge, we screened a library of 618 antiviral compounds to identify agents that could synergize with AmB and enhance its antifungal efficacy. An initial screen identified 18 compounds that enhanced the antifungal activity of AmB. From this hit set, we prioritized the two FDA-approved HIV drugs, cobicistat and elvitegravir, as promising lead candidates. When combined with AmB, both compounds exhibited potent synergistic activity against *A. fumigatus* and other clinically relevant *Aspergillus* species, with FICI values <0.5 in over 90% of isolates tested. To further evaluate the breadth of this synergy, the assay was extended to include *A. brasiliensis, A. flavus, A. niger*, and *A. terreus*. Synergistic interactions were observed in three of the four species tested, while the combination displayed indifference against *A. flavus*. In time-kill assays, both combinations demonstrated sustained fungistatic effects over 48 hours and significantly impaired hyphal development as early as 16 hours, indicating early disruption of fungal growth. Additionally, both cobicistat and elvitegravir significantly enhanced the antibiofilm activity of AmB, reducing biofilm biomass by over 60% when combined with sub-inhibitory AmB concentrations. These combinations also disrupted mature biofilms, achieving up to 80% eradication, a substantial improvement over AmB alone. These findings highlight the potential of these drug combinations as promising treatment options for aspergillosis, leveraging already approved therapies.

**Data availability statement:** All relevant data are within the paper and its Supporting Information files.

**Funding:** This work was supported by the National Institute of Health Grant R01AI141439. The funders had no role in study design, data collection and analysis, decision to publish, or preparation of the manuscript.

**Competing interests:** The authors have declared that no competing interests exist.

## Introduction

Fungi are ubiquitous, ecologically diverse organisms that inhabit virtually every environmental niche [1]. While most coexist harmlessly with humans as commensals or environmental saprophytes, a subset has evolved to act as opportunistic pathogens, particularly in individuals with weakened immune defenses [2,3]. Globally, pathogenic fungi are responsible for a substantial and growing burden of disease, with invasive fungal infections estimated to affect more than 6.5 million people annually [3,4]. These infections account for over 1.6 million deaths each year, a toll comparable to tuberculosis and more than triple the mortality caused by malaria [5]. In U.S. hospitals, fungal pathogens rank among the leading causes of healthcare-associated infections, surpassed only by staphylococci and enterococci [3].

Among these fungal pathogens, *Aspergillus* species, particularly *A. fumigatus*, are major contributors to morbidity and mortality in immunocompromised and critically ill patients [6]. *A. fumigatus* is the principal etiologic agent of invasive aspergillosis (IA), a rapidly progressive and often fatal infection whose incidence has risen alongside the expanding population of immunosuppressed individuals [7]. High-risk groups include patients in intensive care units (ICUs), individuals with chronic obstructive pulmonary disease (COPD), those recovering from surgery, people living with HIV, and recipients of immunosuppressive therapies such as tumor necrosis factor-alpha (TNF-α) inhibitors [8]. In these vulnerable populations, mortality rates for IA have been reported to range from 40% to as high as 90% [7].

*A. fumigatus* is a filamentous, spore-forming fungus that thrives in diverse ecological settings, including soil, decaying vegetation, indoor air, household dust, and drinking water, making human exposure nearly unavoidable [9]. Its small, hydrophobic conidia are readily aerosolized, allowing for efficient airborne dissemination. Humans are estimated to inhale between 500 and 5,000 *Aspergillus* spores daily [10]. In healthy individuals, these inhaled spores are efficiently eliminated by innate immune defenses. However, in immunocompromised patients, conidia can evade clearance, germinate into filamentous hyphae, and invade pulmonary tissues [11].

Hyphal development plays a central role in disease progression [12]. *A. fumigatus* displays a strong affinity for blood vessels, and as hyphae extend through lung tissue, they breach the endothelial lining of adjacent vasculature [13]. This enables fungal elements to enter the bloodstream, promoting hematogenous dissemination that can lead to thrombosis, hemorrhagic infarction, and infection of distant organs. These events underscore the importance of hyphal growth in the pathogenesis of invasive aspergillosis [14].

Beyond its role in tissue invasion, hyphal development also contributes to the formation of biofilms, which are complex, three-dimensional communities of fungal cells embedded within an extracellular matrix. These structures pose significant clinical challenges due to their resistance to antifungal therapies and evasion of host immune responses [15]. Much like bacterial and yeast biofilms, those formed by *A. fumigatus* create a protective niche that impairs drug penetration and immune clearance. Biofilm-associated antifungal resistance may contribute to therapeutic failure even in cases where isolates appear susceptible in standard in vitro assays [13,16].

Emerging evidence suggests that biofilm formation is one of the most important virulence factors in invasive pulmonary aspergillosis (IPA) and aspergilloma [17,18].

Current treatment options for *A. fumigatus* infections rely heavily on a limited number of antifungal drug classes [19,20]. Azoles, particularly voriconazole and isavuconazole, are the recommended first-line agents due to their broad-spectrum activity and availability in both oral and intravenous formulations. However, their clinical use is often complicated by hepatotoxicity, cardiotoxicity (QT prolongation), and significant drug-drug interactions, especially in patients receiving immunosuppressive therapies [20].

Rising rates of azole-resistant *A. fumigatus* isolates have further complicated treatment, with resistant strains now being detected even in azole-naïve patients, likely due to environmental exposure to agricultural azole fungicides [21]. When azoles are ineffective, contraindicated, or poorly tolerated, amphotericin B (AmB) remains a critical alternative. Although the conventional deoxycholate formulation is limited by nephrotoxicity, the liposomal formulation (LAmB) offers improved tolerability and is preferred in cases of azole resistance or intolerance. However, its broader clinical use is constrained by high cost and the need for close monitoring due to risks of nephrotoxicity and hypokalemia [19]. These toxicities are believed to be dose-dependent, underscoring the value of treatment strategies that reduce the required dose without compromising efficacy [22].

Echinocandins such as caspofungin, micafungin, and anidulafungin have limited activity against *Aspergillus* species and are generally used as part of combination or salvage therapy [19]. While their low toxicity and minimal drug-drug interactions offer clinical advantages, their use is limited by modest efficacy and a lack of robust evidence for monotherapy. Additionally, increased echinocandin use has been associated with the emergence of resistance in *Candida* species, warranting cautious application [23].

Given these therapeutic limitations, there is an urgent need for innovative antifungal strategies. Drug repurposing, which involves identifying new applications for existing approved compounds, offers a cost-effective and time-efficient pathway to restore the antifungal arsenal [24]. Combining repurposed agents with AmB may enhance its efficacy while allowing dose reduction, thereby minimizing toxicity [25].

Antiviral drugs represent a particularly attractive class for antifungal repurposing, due to their well-characterized pharmacokinetics, established safety profiles, and frequent use in immunocompromised populations who are also at risk for fungal infections [26]. Our labs prior work has demonstrated that several HIV protease inhibitors (PIs) possess significant antifungal activity, supporting the rationale for their repurposing against invasive fungal infections. Lopinavir, ritonavir, atazanavir, and saquinavir each demonstrated synergistic activity with azoles such as fluconazole, itraconazole, posaconazole against *C. auris*, leading to reduced fungal burden in both in vitro and in vivo models [27–31]. Additionally, several HIV protease inhibitors significantly enhanced the antifungal activity of AmB, exhibiting fungicidal effects, inhibiting biofilm formation, and suppressing hyphal development [32]. In *Cryptococcus*, HIV PIs enhanced the activity of AmB, achieving fungicidal effects and extended post-antifungal activity, with no added cytotoxicity [33]. Beyond yeasts, we recently reported that lopinavir restored susceptibility of *A. fumigatus* to azoles, including itraconazole and posaconazole, via efflux pump inhibition and morphological disruption [34]. Together, these findings highlight the broad-spectrum antifungal potential of HIV protease inhibitors and underscore the need to explore additional classes of antiviral agents in combination with AmB, particularly against filamentous fungi such as *A. fumigatus*.

In this study, we screened a library of 618 approved and investigational antiviral compounds for their ability to enhance the antifungal activity of amphotericin B (AmB) against *Aspergillus fumigatus* strain AF293. Screening libraries of compounds provides an efficient way to uncover drugs that may enhance or interfere with existing antifungals, guiding the development of effective combination therapies [35]. From this screen, we identified two FDA-approved HIV drugs, cobicistat (COB) and elvitegravir (ELV), which are co-formulated in clinical use [36,37]. These agents were prioritized for further evaluation based on their established safety profiles, clinical relevance, and routine use in people living with HIV, a population at heightened risk for invasive aspergillosis [37]. Their effects were subsequently assessed across multiple

*Aspergillus* species, with additional investigations into their impact on key fungal virulence traits, including hyphal development and biofilm formation.

## Materials and methods

### Fungal strains, media and chemicals

Fungal strains used in this study (listed in Table 1) were obtained from the Centers for Disease Control and Prevention (CDC, Atlanta, GA) and the American Type Culture Collection (ATCC). RPMI 1640 medium was prepared using components from Thermo Fisher Scientific (Waltham, MA) and buffered with 3-(N-Morpholino) propane sulfonic acid (MOPS; Sigma-Aldrich, St. Louis, MO). Potato Dextrose (PD) broth and agar were purchased from Becton, Dickinson and Company (Franklin Lakes, NJ). The Antiviral Compound Library (Catalog No. HY-L027) was sourced from MedChemExpress (Monmouth Junction, NJ). Amphotericin B (AmB) was obtained from Chem-Impex International (Wood Dale, IL), while the test compounds elvitegravir (ELV) and cobicistat (COB) were purchased from Ambeed (Arlington Heights, IL) and Astatech (Bristol, PA), respectively. Crystal violet and phosphate-buffered saline (PBS) were purchased from Acros Organics (NJ, USA) and Corning (VA, USA), respectively.

### Screening of the antiviral compound library and identification of compounds

The MedChemExpress (MCE) Antiviral Compound Library, consisting of 618 agents, was screened in vitro to identify compounds that enhance the antifungal activity of sub-inhibitory amphotericin B (AmB) against *A. fumigatus*. Each compound was tested in RPMI medium containing AmB at 0.25 μg/mL, which corresponds to 0.125 × the minimum inhibitory concentration (MIC), using the reference strain *A. fumigatus* AF293 (ATCC MYA-4609). Compounds were dispensed into 96-well

**Table 1. Summary of hit compounds identified from the antiviral library screen demonstrating enhanced antifungal activity in combination with amphotericin B.**

| Compound | Growth Inhibition (%)# | Description |
|---|---|---|
| Tunicamycin | 88.6 | Inhibits N-linked glycosylation |
| TAK779 | 92.1 | Inhibitor of CCR5 and CXCR3 |
| MSC1094308 | 91.2 | Reversible VPS4B/p97 (VCP) (I/II type AAA ATPase) allosteric inhibitor |
| Tubacin | 80.6 | Selective inhibitor of HDAC6 |
| Miltefosine | 91.3 | Inhibits PI3K/Akt activity |
| Cobicistat | 85.1 | Selective inhibitor of cytochrome P450 3A |
| KW8232 | 89.0 | Reduces the biosynthesis of PGE2 |
| NH125 | 92.2 | Selective inhibitor of eukaryotic elongation factor 2 kinase |
| Auranofin | 89.8 | Inhibitor of thioredoxin reductase (TrxR) |
| Cetylpyridinium | 91.9 | Anti-HBV capsid assembly inhibitor |
| Pritelivir | 86.9 | Inhibitor of the viral helicase-primase complex |
| Pirodavir | 92.0 | Broad-spectrum picornavirus inhibitor |
| Elvitegravir | 85.7 | HIV integrase inhibitor |
| Amenamivir | 84.6 | Helicase-primase inhibitor |
| Dapivirine | 86.8 | Nonnucleoside reverse transcriptase inhibitor (NRTI) |
| DBEQ | 81.5 | Potent, reversible, and ATP-competitive p97 inhibitor |
| Acriflavine | 84.7 | Cancer research agent and potent HIF-1 inhibitor |
| Saquinavir | 86.9 | HIV protease inhibitor |

#Percent growth inhibition values represent the reduction in *A. fumigatus* biomass after 48 hours of incubation, relative to the untreated growth control.

plates at a final concentration of 16 µM, with the first and last columns serving as growth controls without any drug. Each well received 100 µL of AmB-supplemented RPMI medium containing $1 \times 10^4$ conidia. Plates were then incubated at 35 °C for 48 hours, and fungal growth was evaluated both visually and by measuring optical density at 530 nm using a Tecan F200 Pro Multi-Mode Plate Reader to identify compounds that enhanced AmB activity.

## Microdilution checkerboard assays

To evaluate the activity of cobicistat and elvitegravir against a range of *Aspergillus* species, we performed standard broth microdilution checkerboard assays as previously described for filamentous fungi [34,38]. MIC and checkerboard readings were recorded at 48 hours post-incubation, with wells showing nearly complete visual growth inhibition considered for analysis. Fractional inhibitory concentration indices (ΣFICIs) were calculated to assess drug interactions. Interactions were classified as synergistic (SYN) when ΣFICI ≤ 0.5, indifferent (IND) when >0.5 to ≤4, and antagonistic (ANT) when >4 [39]. Checkerboard assays were performed in two independent biological experiments using two freshly prepared separate cultures.

## Time kill assay

To evaluate the impact of treatment combinations on the growth kinetics of *A. fumigatus*, a time-kill assay was performed as previously described for filamentous fungi [34]. Fungal spores were diluted in RPMI to a final concentration of $5 \times 10^4$ conidia/mL and exposed to the combination treatments AmB/COB and AmB/ELV.

   Amphotericin B alone at 2 µg/mL served as positive control, while untreated wells were used as negative controls. Wells containing COB and ELV alone were used to assess whether they had any effect on fungal growth. Treatments included AmB at two sub-inhibitory concentrations (0.25 and 0.5 µg/mL), tested in combination with COB or ELV at fixed concentrations of 2 µg/mL and 4 µg/mL. Specifically, AmB 0.25 µg/mL was combined with COB or ELV at 4 µg/mL, while AmB 0.5 µg/mL was combined with both 2 µg/mL and 4 µg/mL of COB or ELV to assess dose-dependent effects on fungal growth. Fungal growth was monitored by measuring optical density at 530 nm ($OD_{530}$) using a Tecan F200 Pro Multi-Mode Plate Reader over a period of 48 hours. Each treatment was tested in five independent wells in two independent biological experiments using individual cultures.

## Hyphal growth assay

To evaluate the effect of AmB/COB and AmB/ELV combinations on hyphal development, *A. fumigatus* AF293 conidia ($5 \times 10^4$ conidia/mL) were inoculated into RPMI-1640 medium containing synergistic concentrations of each drug combination, as previously described [40]. Cultures were incubated at 37 °C for 16 hours. Following incubation, fungal morphology was examined using a Nikon Eclipse Ti2 inverted microscope at 20 × magnification, and representative brightfield images were acquired. Hyphal lengths were measured using ImageJ software, utilizing the measure tool, with at least 15 hyphae quantified from two independent visual fields per condition to assess differences in filamentous growth. Experiment was performed in two independent biological experiments.

## Biofilm inhibition assay

The effect of antiretroviral combinations on the prevention of biofilm formation was assessed as previously described, using two independent biological replicates, each performed in technical duplicate. [28,32,41]. Wells containing only fungal culture (no drug) served as the negative control. Wells with COB or ELV alone were included to assess their individual effects on biofilm inhibition. COB and ELV were added at fixed concentrations (16 µg/mL), while AmB was serially diluted in RPMI 1640 medium. The drugs were added to 96-well plates containing approximately $1 \times 10^4$ CFU/mL of *A. fumigatus* spores. Plates were incubated at 35 °C for 24 hours to allow for biofilm development. Following incubation, wells were stained with 100 µL of 0.1% (w/v) crystal violet for 30 minutes. Excess stain was removed, and wells were washed three times with 200 µL of distilled water, then air-dried. Biofilm biomass was quantified by solubilizing the retained crystal violet in 100 µL of 70% ethanol, and absorbance was measured at 600 nm ($OD_{600}$).

## Biofilm eradication assay

A biofilm eradication assay was performed, with modifications to a previously described protocol, to evaluate the effect of antiretroviral combinations with amphotericin B on preformed *A. fumigatus* biofilms [41]. Experiments were done twice using separate cultures, with each experiment run in technical duplicates. Briefly, fungal suspensions ($1 \times 10^6$ CFU/mL) were dispensed into flat-bottom 96-well tissue culture-treated plates and incubated at 37 °C for 24 hours to allow biofilm formation. After incubation, the supernatant was carefully removed, and wells were washed twice with sterile phosphate-buffered saline (PBS) to eliminate non-adherent cells. Preformed biofilms were treated with serial dilutions of AmB in combination with fixed concentrations of COB or ELV, prepared in RPMI-1640 medium supplemented with MOPS. Wells containing only fungal biofilm (no drug) served as the negative control. Additional wells containing AmB, COB, or ELV alone were included to assess the individual effects of each compound and compare them to the combination treatments. Plates were incubated again at 37 °C for 24 hours. Following treatment, wells were washed twice with PBS to remove residual drug and non-adherent cells. Biofilm viability was quantified using the XTT [2,3-bis(2-methoxy-4-nitro-5-sulfophenyl)-2H-tetrazolium-5-carboxanilide] reduction assay. A freshly prepared XTT-menadione solution was added to each well and incubated in the dark at 37 °C for 2–3 hours. Colorimetric changes were measured at 492 nm using a microplate reader to assess metabolic activity of the biofilm-embedded cells.

## Statistical analysis

Data were analyzed using a one-way ANOVA, with post hoc Dunnet's multiple comparisons test ($P < 0.05$) (GraphPad Software, La Jolla, CA).

## Results

### Screening of antiviral drug library identifies potent AmB adjuvant compounds (Cobicistat and Elvitegravir)

To identify compounds capable of enhancing the antifungal activity of AmB, we screened the Antiviral Compound Library from MedChemExpress (MCE; Cat. No. HY-L027), which consists of 618 investigational and approved compounds, at a fixed concentration of 16 µM against the *Aspergillus fumigatus* reference strain AF293. Compounds were tested in the presence or absence of sub-inhibitory concentration of AmB (0.25 µg/mL equivalent to 0.125x MIC), a concentration selected to minimize background inhibition while allowing for identification of potentiating agents. The assay was performed in RPMI 1640 medium, and fungal growth was assessed after 48 hours. Compounds were considered positive hits if they inhibited fungal growth by ≥80% only in the presence of AmB. Initial hits were identified by visual inspection and further confirmed by spectrophotometric measurement at optical density ($OD_{530}$).

As per our criteria, a total of 18 compounds (2.91% hit rate) demonstrated enhanced antifungal activity in combination with AmB (Table 1). These included 9 antiviral agents (primarily targeting HIV and HSV), 7 anticancer drugs and 2 antiseptic compounds.

From this group, two FDA-approved HIV antiretrovirals (HIV-ARTs), cobicistat (COB) and elvitegravir (ELV), were selected for further investigation against *A. fumigatus* (Fig 1). These compounds were selected based on their clinical relevance, established safety profiles, and existing use in populations at high risk for invasive fungal infections, such as individuals with HIV/AIDS. Their regulatory approval and widespread clinical use reduce the translational barrier for repurposing, potentially accelerating the path toward clinical application for fungal infections caused by *A. fumigatus*.

### In-vitro synergy of antiretroviral combinations with AmB against *A. fumigatus*

To assess the potential for synergistic interactions between AmB and the antiretroviral agents COB and ELV, checker-board assays were performed against a panel of *A. fumigatus* isolates. Both combinations exhibited strong synergy across

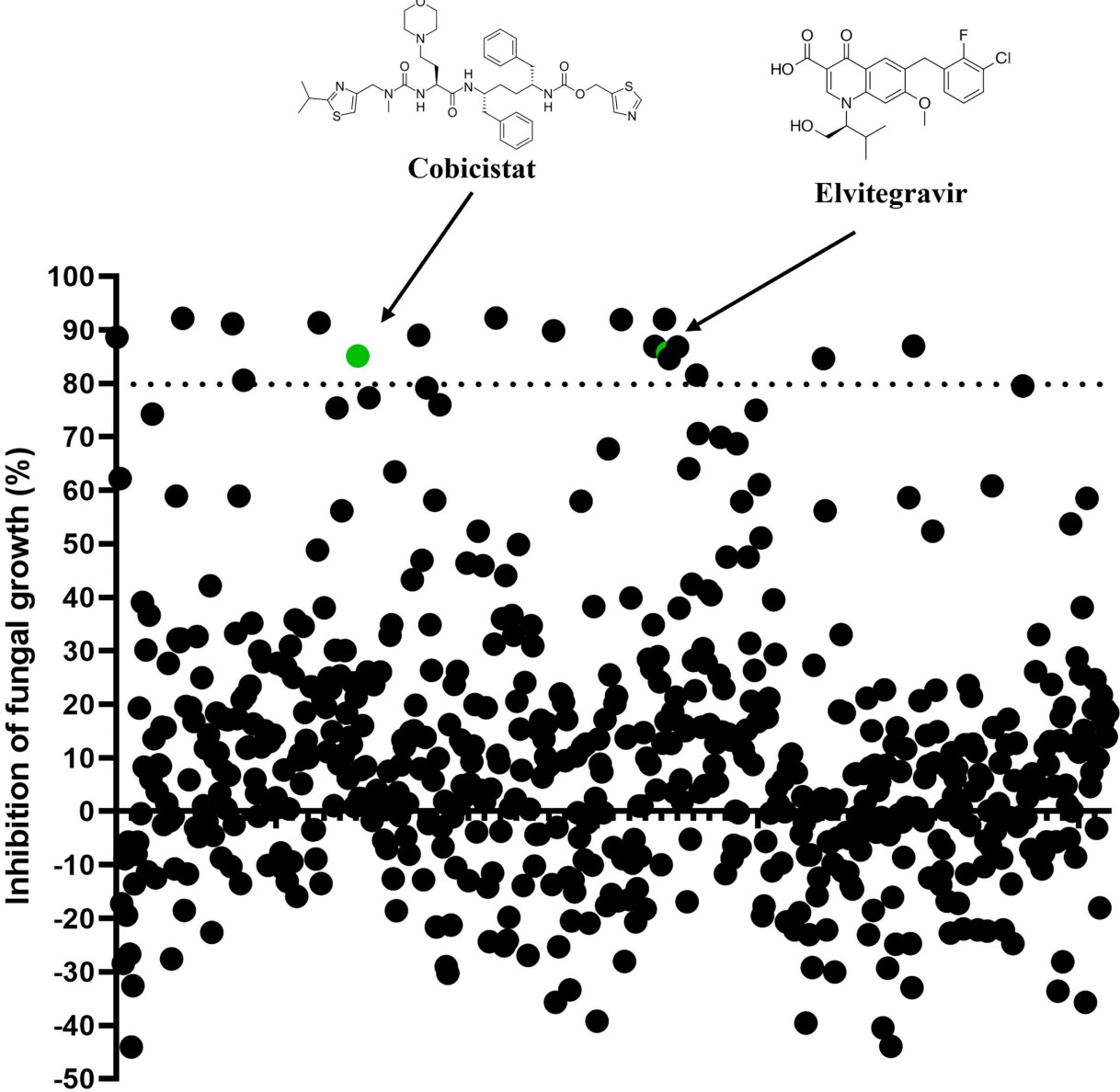

**Fig. 1. Identification of antiviral compounds with amphotericin B potentiating activity against *A. fumigatus*.** Screening of the MedChemExpress (MCE) Antiviral Compound Library at a final concentration of 16 µM against *A. fumigatus* AF293 in the presence of a subinhibitory concentration of AmB (0.25 µg/mL). Fungal growth was assessed after 48 hours by measuring the optical density at 530 nm. Green dots indicate compounds that achieved ≥80% growth inhibition compared to the no-drug control. The chemical structures of the selected hit compounds, cobicistat (COB) and elvitegravir (ELV), are shown above the screening plot.

all 11 *A. fumigatus* strains tested, with ΣFICI values ranging from 0.13 to 0.26 for AmB/COB and 0.08 to 0.27 for AmB/ELV (Table 2). To further evaluate the breadth of this synergy, the assay was extended to include other clinically important *Aspergillus* species, including *A. brasiliensis*, *A. flavus*, *A. niger*, and *A. terreus*. Synergistic interactions were observed in three of the four *Aspergillus* species tested, while the combination displayed indifference against *A. flavus*, supporting the broader antifungal potential of these drug combinations across multiple species.

**Table 2. Interaction between amphotericin B (AmB) with cobicistat (COB) and elvitegravir (ELV) against *Aspergillus* isolates.**

| Strain | Data for AmB/COB | | | | Data for AmB/ELV | | | |
|---|---|---|---|---|---|---|---|---|
| | MIC (µg/mL) | | ΣFICI | Mode | MIC (µg/mL) | | ΣFICI | Mode |
| | Alone | Combined | | | Alone | Combined | | |
| AF293 | 2/>128 | 0.25/1 | 0.133 | SYN | 2/>128 | 0.25/4 | 0.156 | SYN |
| CDC731 | 1/>128 | 0.25/1 | 0.258 | SYN | 2/>128 | 0.125/2 | 0.078 | SYN |
| CDC732 | 2/>128 | 0.25/2 | 0.141 | SYN | 2/>128 | 0.25/2 | 0.141 | SYN |
| CDC733 | 2/>128 | 0.25/4 | 0.156 | SYN | 2/>128 | 0.25/2 | 0.141 | SYN |
| CDC734 | 1/>128 | 0.25/1 | 0.258 | SYN | 0.5/>128 | 0.06/2 | 0.136 | SYN |
| CDC735 | 2/>128 | 0.25/2 | 0.141 | SYN | 2/>128 | 0.125/4 | 0.094 | SYN |
| CDC736 | 2/>128 | 0.25/2 | 0.141 | SYN | 2/>128 | 0.125/4 | 0.094 | SYN |
| CDC737 | 2/>128 | 0.25/2 | 0.141 | SYN | 2/>128 | 0.25/2 | 0.141 | SYN |
| CDC738 | 2/>128 | 0.25/4 | 0.156 | SYN | 2/>128 | 0.5/2 | 0.266 | SYN |
| CDC739 | 2/>128 | 0.25/4 | 0.156 | SYN | 2/>128 | 0.25/2 | 0.141 | SYN |
| CDC740 | 2/>128 | 0.5/1 | 0.258 | SYN | 2/>128 | 0.25/2 | 0.141 | SYN |
| *A. brasiliensis* CBS 733.88 | 1/>128 | 0.125/4 | 0.156 | SYN | 1/>128 | 0.125/2 | 0.141 | SYN |
| *A. niger* 6275 CBS 131.52 | 2/>128 | 0.5/1 | 0.258 | SYN | 2/>128 | 0.5/0.25 | 0.266 | SYN |
| *A. terreus* 1012 ATCC 10071 | 2/>128 | 0.25/4 | 0.156 | SYN | 2/>128 | 0.5/0.25 | 0.266 | SYN |
| *A. flavus* 9643 CBS 131.61 | 2/>128 | 2/1 | 1.008 | IND | 2/>128 | 2/1 | 1.008 | IND |

Fractional inhibitory concentration indices (ΣFICIs) were calculated to assess drug interactions. Interactions were classified as synergistic (SYN) when ΣFICI ≤ 0.5, indifferent (IND) when >0.5 to ≤4, and antagonistic (ANT) when >4

## HIV antiretrovirals potentiate AmB to suppress *A. fumigatus* growth

To investigate the killing kinetics of AmB in combination with either COB or ELV against *A. fumigatus* AF293, time-kill assays were performed using two sub-inhibitory concentrations of AmB (0.25 and 0.5 µg/mL) in combination with fixed concentrations (2 and 4 µg/mL) of the antiretrovirals. As expected, AmB alone at 0.25 or 0.5 µg/mL, as well as COB or ELV alone (at 4 µg/mL), did not significantly inhibit growth relative to the untreated control (Fig 2A). However, when AmB 0.25 µg/mL was combined with COB or ELV at 4 µg/mL, temporary inhibition of growth was observed up to 24 hours, after which growth resumed (Fig 2B and 2C). A similar pattern was seen for combinations of AmB 0.5 µg/mL with COB or ELV

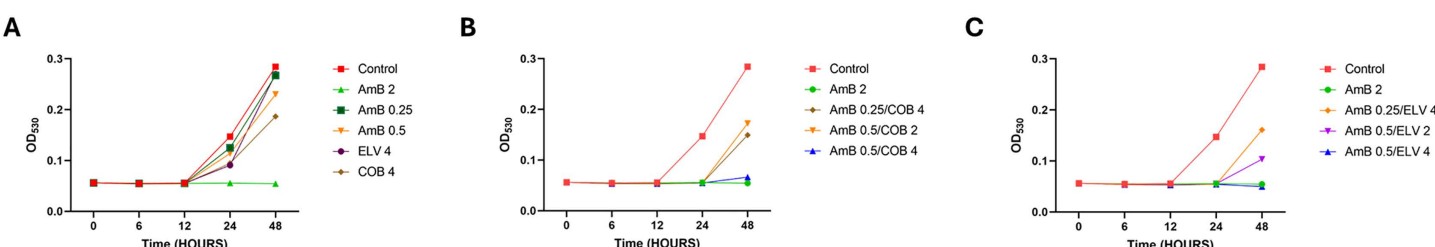

**Fig 2. Effect of antiretrovirals alone and in combination with amphotericin B on the growth kinetics of *A. fumigatus* AF293.** *A. fumigatus* AF293 conidia ($5 \times 10^4$ conidia/mL) were incubated in RPMI-MOPS medium and fungal growth was monitored over a 48-hour period by measuring optical density at 530 nm at regular time intervals. **(A)** Growth kinetics of untreated control versus monotherapies: AmB at 0.25, 0.5, and 2 µg/mL; COB and ELV at 4 µg/mL. Only AmB at 2 µg/mL resulted in substantial inhibition, while sub-inhibitory doses and antiretrovirals alone had minimal impact. **(B)** Comparison of COB combinations: AmB 0.25/COB 4, AmB 0.5/COB 2, and AmB 0.5/COB 4 µg/mL. All combinations showed enhanced inhibition versus monotherapies, with the 0.5/4 pairing sustaining growth suppression across 48 hours. (C) Comparison of ELV combinations: AmB 0.25/ELV 4, AmB 0.5/ELV 2, and AmB 0.5/ELV 4 µg/mL. As with COB, the highest dose pairing (0.5/4) achieved the most durable inhibition.

at 2 µg/mL. Notably, the most sustained growth suppression was achieved when AmB 0.5 µg/mL was paired with COB or ELV at 4 µg/mL, with fungal inhibition maintained across the full 48-hour time course.

## Antiretroviral combinations inhibit hyphal development of *A. fumigatus*

To further evaluate the effect of antiretroviral–AmB combinations on fungal morphology, *A. fumigatus* conidia were incubated for 16 hours in RPMI medium under each treatment condition. Microscopic analysis revealed that hyphal growth was significantly reduced in both AmB/COB and AmB/ELV groups compared to untreated controls and monotherapies (Fig 3A). Quantitative measurement of hyphal lengths confirmed this observation, with combination treatments showing significantly less growth compared to the untreated control. (Fig 3B). These results indicate that the synergistic combinations impair early hyphal development, a key process in fungal invasion and pathogenesis.

## Anti-retroviral combination inhibits biofilm formation

Biofilm formation is a key virulence factor in invasive aspergillosis, contributing to antifungal resistance and persistence in host tissues. To evaluate the impact of the antiretroviral combinations on this pathogenic trait, *A. fumigatus* AF293 was incubated for 24 hours in the presence of COB (16 µg/mL) or ELV (16 µg/mL), each combined with a sub-inhibitory concentration of AmB (0.015 µg/mL, equivalent to $0.0075 \times MIC$). Total biofilm biomass was then quantified using crystal violet staining. Both combinations significantly reduced biofilm formation compared to the untreated controls. Specifically, AmB/COB reduced biofilm mass by ~72%, while AmB/ELV resulted in a ~62% reduction (Fig 4). These effects were statistically significant ($p < 0.0001$ for both combinations vs. untreated control). In comparison, AmB alone at subinhibitory concentrations and COB or ELV alone did not significantly reduce the biofilm formation.

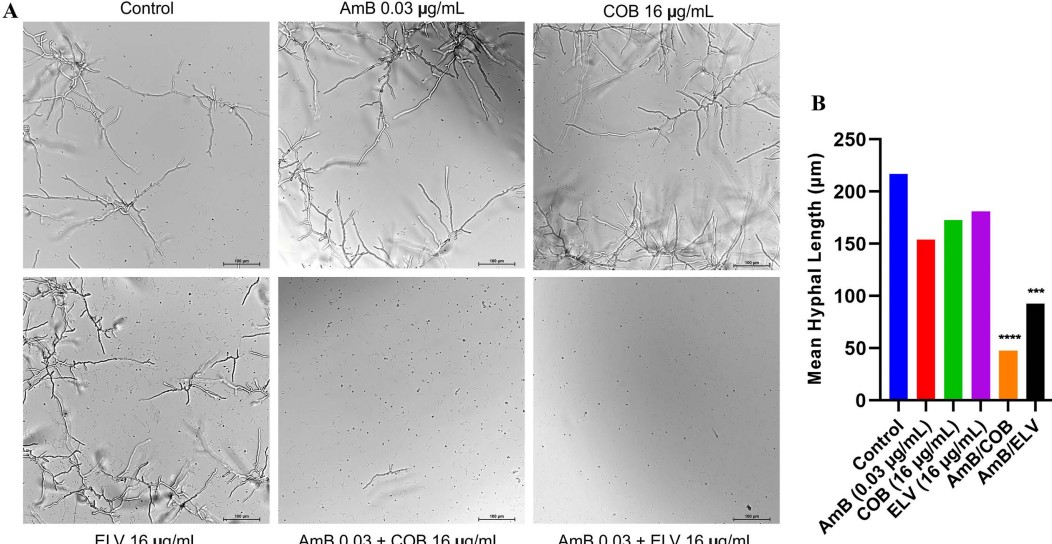

**Fig 3. Inhibition of *A. fumigatus* hyphal growth by antiretroviral–amphotericin B combinations. (A)** Brightfield microscopy images of *A. fumigatus* AF293 grown for 16 h in RPMI-1640 medium under the following conditions: untreated control, amphotericin B (AmB, 0.03 µg/mL), cobicistat (COB, 16 µg/mL), elvitegravir (ELV, 16 µg/mL), AmB/COB, and AmB/ELV. Images were acquired at 20× magnification on a Nikon Eclipse Ti2 microscope, with a 100 µm scale bar shown. **(B)** Quantification of hyphal lengths under each condition was done using ImageJ software. For each variable, 15 hyphae were measured, and data are presented as mean. Statistical significance was assessed using one-way ANOVA followed by Dunnett's post hoc test for multiple comparisons. Asterisks indicate significant differences relative to the untreated control, *** (P < 0.005) and **** (P < 0.0001).

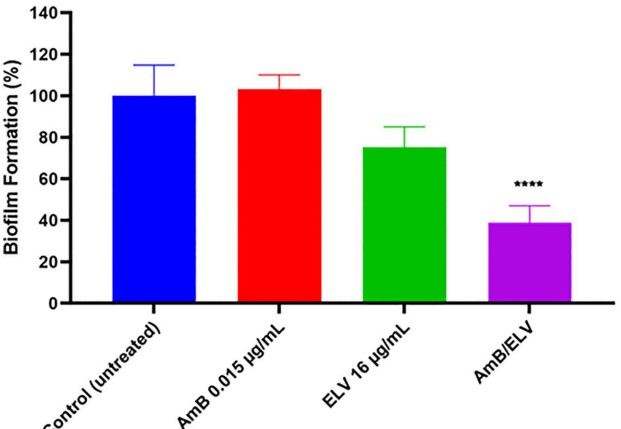
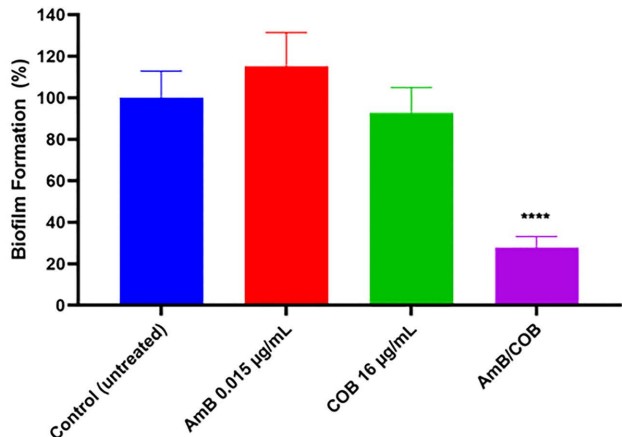

**Fig 4. Inhibition of *A. fumigatus* AF293 biofilm formation by amphotericin B in combination with HIV antiretrovirals.** Biofilms were established by incubating *A. fumigatus* AF293 ($1 \times 10^4$ CFU/mL) in RPMI-1640 medium for 24 hours in the presence of cobicistat or elvitegravir (16 µg/mL), either alone or combined with a sub-inhibitory concentration of amphotericin B (0.015 µg/mL; $0.0075 \times$ MIC). Biofilm biomass was quantified using crystal violet staining, followed by solubilization and absorbance measurement at 600 nm. Statistical analysis was performed using one-way ANOVA with Dunnett's post hoc test. Significant reductions were observed for both AmB/COB and AmB/ELV combinations. Asterisks indicate significant differences relative to the untreated control **** ($p < .0001$).

## Activity against pre-formed biofilms of *Aspergillus* species

Given the clinical relevance of biofilm-associated antifungal resistance in invasive aspergillosis, we further assessed the efficacy of antiretroviral/AmB combinations against established *A. fumigatus* biofilms. One-day-old pre-formed biofilms of strain AF293 were exposed to increasing concentrations of AmB/COB and AmB/ELV, and metabolic activity was quantified using the XTT reduction assay. Both combinations demonstrated robust biofilm-disruptive activity. Specifically, the AmB/COB combination significantly reduced metabolic activity of the biofilm by 85.4%, while the AmB/ELV combination resulted in an 80.6% reduction, relative to untreated control (Fig 5). ($p < 0.0001$ for both combinations vs. untreated control).

## Discussion

Invasive aspergillosis (IA) remains a major contributor to global morbidity and mortality, with an estimated annual incidence exceeding 2 million cases and approximately 1.8 million associated deaths worldwide [4]. Although clinically recognized as early as 1953, IA continues to carry unacceptably high mortality rates, reaching up to 90% in some high-risk patient populations [42,43].

The burden of IA is particularly pronounced among immunocompromised individuals [44]. Classic predisposing factors include prolonged neutropenia, chemotherapy for hematologic malignancies, hematopoietic stem cell transplantation (HSCT), graft-versus-host disease, solid organ transplantation, and long-term corticosteroid therapy. Additional high-risk groups include patients with advanced AIDS, functional neutrophil disorders such as chronic granulomatous disease, and those with viral-associated pulmonary aspergillosis (VAPA) [45].

Despite advancements in diagnostics and antifungal therapies, IA remains a formidable clinical challenge, increasingly compounded by the global rise of azole-resistant *A. fumigatus* strains [19,46]. Current treatment strategies rely on three major antifungal classes, azoles, polyenes, and echinocandins, each with distinct pharmacologic profiles and limitations [19,47]. However, in complex clinical scenarios involving antifungal resistance, intolerance, significant organ dysfunction, or drug–drug interactions, existing therapies often fail to achieve optimal outcomes [19]. These limitations highlight the urgent need for novel and more effective therapeutic approaches [48].

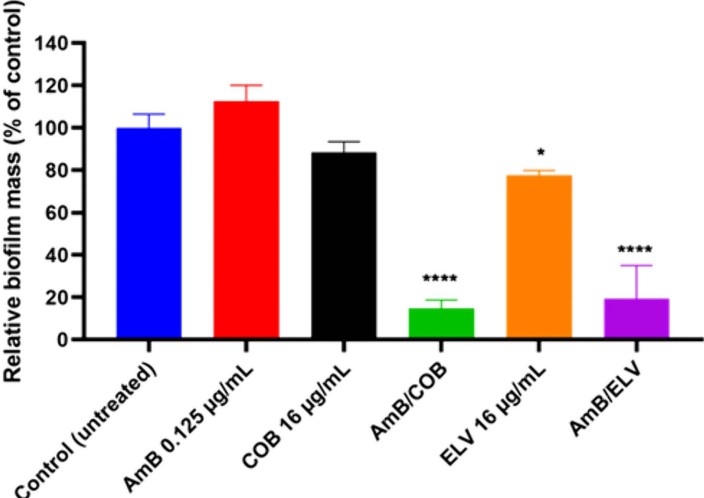

**Fig 5. Disruption of pre-formed *A. fumigatus* biofilms by amphotericin B in combination with cobicistat or elvitegravir.** Mature biofilms of *A. fumigatus* AF293 were allowed to form over 24 hours prior to treatment with increasing concentrations of AmB/COB and AmB/ELV combinations. Following 24 hours of drug exposure, residual metabolic activity was quantified using the XTT reduction assay. Statistical analysis was performed using one-way ANOVA with Dunnett's test. Significant reductions in metabolic activity were observed for AmB/COB and AmB/ELV. Asterisks indicate significant differences relative to the untreated control. * ($p$ = 0.0458) and **** ($p$ < 0.0001).

Combination antifungal therapy (CAF) has gained attention as a strategy to improve treatment efficacy and reduce toxicity in invasive fungal infections. This approach offers several advantages, including synergistic fungicidal activity, broader antimicrobial coverage during empirical treatment, and a reduced risk of resistance development [49,50]. Notably, CAF enables dose reduction of toxic agents such as amphotericin B (AmB), helping to minimize adverse effects like nephrotoxicity, while maintaining or enhancing therapeutic efficacy [49].

In the treatment of *Aspergillus* infections, especially those caused by *A. fumigatus*, combination regimens have shown promising outcomes [50]. Findings from a large international randomized controlled trial suggest that combining voriconazole with anidulafungin may improve six-week survival in patients with hematologic malignancies and invasive aspergillosis [51].

Another complementary strategy is drug repurposing, which accelerates development and reduces costs by leveraging existing pharmacologic and safety data [52]. This approach involves screening libraries of approved or investigational compounds that were originally developed for non-fungal targets to uncover unexpected anti-fungal activity. Notably, such efforts have led to the identification of clofazimine, an antimycobacterial agent, as a synergistic partner with caspofungin and posaconazole against *A. fumigatus* [53]. Similarly, the HIV protease inhibitor lopinavir has demonstrated potential as an adjunctive therapy, particularly in addressing azole-resistant *Aspergillus* strains [34].

In this study, we applied a dual strategy combining drug repurposing and combination therapy to address the limitations of current antifungal treatments. Given their clinical relevance, we screened a library of antiviral compounds for synergistic interactions with AmB against *A. fumigatus*. From this screen, two antiretroviral agents, cobicistat and elvitegravir, emerged as promising candidates and were selected for further evaluation.

Elvitegravir and cobicistat are co-formulated in the fixed-dose HIV therapy Genvoya®, which also includes emtricitabine and tenofovir alafenamide [37]. Elvitegravir is an HIV integrase inhibitor administered once daily, typically paired with a pharmacokinetic booster like cobicistat to increase systemic exposure, and has a predictable half-life of approximately 10 hours [54]. Cobicistat, a selective CYP3A4 inhibitor approved in 2012, lacks intrinsic antiviral activity but has demonstrated efficacy and tolerability as a pharmacoenhancer in Phase II and III trials [55,56]. Notably, this combination has

shown good safety and tolerability in HIV-infected adults with end-stage renal disease on hemodialysis, with no serious renal-related adverse events reported and only mild side effects observed [57].

In our study, both compounds showed robust synergy with AmB across a panel of clinical *Aspergillus* isolates. Checkerboard assays revealed synergy and was further confirmed by time-kill assays that showed that AmB at 0.5 μg/mL in combination with 4 μg/mL COB or ELV closely mirrored the inhibitory effect of a higher dose of AmB dose over 48 hours.

In addition to fungistatic effects, we observed significant disruption of fungal virulence traits. Both combinations impaired hyphal elongation, a critical step in host tissue invasion, and substantially reduced biofilm formation by ~60–70%. Furthermore, they exhibited potent activity against established biofilms, reducing metabolic activity in mature structures by over 80%. Targeting hyphal growth is pivotal in combating *A. fumigatus* infections because the germination of inhaled conidia into filamentous hyphae represents a critical early step in host tissue invasion and is essential for disease establishment [58,59]. Moreover, hyphal proliferation is the foundation for biofilm development which is particularly relevant given that *A. fumigatus* forms robust biofilms during both acute and chronic infections [17,60]. Biofilms not only serve as a physical barrier against host immune responses but also significantly diminish antifungal drug penetration and efficacy [61]. This microenvironment contributes to treatment failure even in cases where isolates are deemed susceptible by standard in vitro susceptibility testing. Indeed, persistent high mortality rates in invasive aspergillosis, despite antifungal therapy, may be attributed in part to biofilm-mediated resistance [17,62].

Since elvitegravir and cobicistat are already safely co-formulated, combining them with AmB presents a viable opportunity for a three-drug antifungal regimen. Such a combination could potentially improve in vivo efficacy, especially in patients with biofilm-associated disease, while minimizing AmB-associated toxicity through dose reduction.

Our findings demonstrate that antiretroviral agents, particularly cobicistat and elvitegravir, synergize effectively with amphotericin B (AmB) against *A. fumigatus* in vitro, supporting their potential as adjunctive agents in polyene-based antifungal therapy. This strategy builds on previous successes in treating other invasive fungal infections through combination approaches. For example, the pairing of AmB with flucytosine has been shown to significantly improve survival in patients with cryptococcal meningitis [63]. Similarly, combination therapies involving echinocandins, as well as HIV protease inhibitors as previously reported by our group, have demonstrated efficacy against *Candida auris* isolates [32,64].

Our results are distinctive in repurposing HIV antivirals, already in clinical use for patients at elevated risk of aspergillosis, to enhance AmB efficacy. The demonstrated synergy, alongside suppression of virulence traits such as hyphal growth and biofilm formation, suggests a practical path for therapeutic intervention. To take these results from the lab to the clinic, the next steps should include mechanistic studies and in vivo validation, ideally using models such as *Galleria mellonella* or murine invasive aspergillosis to confirm safety and efficacy.

## Supporting information

**S1 Data. In-vitro activity.**
(XLSX)

## Acknowledgments

We acknowledge the CDC and ATCC for providing the fungal isolates used in this study.

## Author contributions

**Conceptualization:** Ammar A Khan.

**Data curation:** Ammar A Khan.

**Formal analysis:** Ammar A Khan.

**Funding acquisition:** Mohamed N. Seleem.

**Methodology:** Ammar A Khan, Ehab A. Salama.

**Project administration:** Mohamed N. Seleem.

**Resources:** Mohamed N. Seleem.

**Supervision:** Mohamed N. Seleem.

**Writing – original draft:** Ammar A Khan.

**Writing – review & editing:** Ammar A Khan, Ehab A. Salama, Mohamed N. Seleem.

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
