## [Decision Letter · Decision Letter 0]

6 Oct 2025

PONE-D-25-49174Synergistic antifungal activity of antiretrovirals with amphotericin B against Aspergillus species.PLOS ONE

Dear Dr. Seleem,

Thank you for submitting your manuscript to PLOS ONE. After careful consideration, we feel that it has merit but does not fully meet PLOS ONE’s publication criteria as it currently stands. Therefore, we invite you to submit a revised version of the manuscript that addresses the points raised during the review process.

We look forward to receiving your revised manuscript.

Kind regards,

Aijaz Ahmad, Ph.D.

Academic Editor

PLOS ONE

“This work was supported by the National Institute of Health Grant R01AI141439.”

3. We note that your Data Availability Statement is currently as follows: [All relevant data are within the manuscript and its Supporting Information files]

Reviewers' comments:

Reviewer's Responses to Questions

**Comments to the Author**

1. Is the manuscript technically sound, and do the data support the conclusions?

Reviewer #1: Yes

Reviewer #2: Yes

Reviewer #3: Yes

2. Has the statistical analysis been performed appropriately and rigorously?

Reviewer #1: Yes

Reviewer #2: No

Reviewer #3: Yes

3. Have the authors made all data underlying the findings in their manuscript fully available?

Reviewer #1: Yes

Reviewer #2: No

Reviewer #3: Yes

4. Is the manuscript presented in an intelligible fashion and written in standard English?

Reviewer #1: Yes

Reviewer #2: Yes

Reviewer #3: Yes

5. Review Comments to the Author

Reviewer #1: My suggestions and comments to authors are as follows

1) Minor English editing is needed

2) Author can add imp. reference where re screening of drugs was performed (see https://journals.asm.org/doi/10.1128/aac.00484-25)

3) Author can include main target of antiviral compound listed in table 1

4) Quality of Fig 3 panel A images can be improved (better contrast between background and hypha (can be zoom in if needed)

5) In Fig 3 panel A , mark scale bar properly

6) Better to check toxicity on human cell line (with same combination as drugs as in fig 4, 5

Reviewer #2: - Major Concerns

1) Number of Experimental Replicates

The manuscript lacks clarity regarding the number of biological versus technical replicates. For example:

The time-kill assay reports five independent wells per treatment, but it is unclear whether these represent biological or only technical replicates. Best practice would require at least two independent biological experiments, each with triplicates.

Checkerboard assays were only performed in duplicate, which seems insufficient for robust reproducibility.

I recommend explicitly stating whether each experiment was performed with at least two independent biological repeats (using separate cultures) and providing clear replication numbers across all assays.

2) Hyphal Growth Assay – Methodological Detail

The description of the hyphal growth assay is incomplete:

The authors mention measuring hyphal lengths with ImageJ, but they do not specify the plugin or tool used (e.g., Simple Neurite Tracer, Measure, Analyze Skeleton). Reproducibility requires reporting the exact ImageJ plugin/workflow.

Furthermore, while images are provided (e.g., Fig. 3), the scale bar is not visible in the main text figures, and while magnification (20×) is reported.

3) Time-Kill Assay – Controls

The assay includes amphotericin B alone as a positive control and untreated wells as negative controls. However:

The manuscript itself states in the results that COB and ELV alone did not significantly inhibit fungal growth, suggesting they were tested.

Still, the methods section is inconsistent—it does not explicitly describe inclusion of single-drug controls for COB and ELV, creating ambiguity.

For a proper synergy evaluation, single-drug controls should be explicitly included and consistently described in both methods and results.

- Minor Concerns

1) Terminology Consistency

The text alternates between “antiretroviral combinations” and “HIV-ARTs.” Consistent terminology would improve readability.

2) Controls in Biofilm Assays

The biofilm inhibition and eradication assays do not clearly describe inclusion of single-drug controls (COB alone, ELV alone, AmB alone). This omission makes it difficult to interpret whether synergy is real or if effects derive from additive actions.

3) Data Transparency

Figures (e.g., Figs. 2–5) report mean values, but no raw data availability (e.g., per well OD values, per hypha measurements) is indicated beyond the blanket statement “all relevant data are within the manuscript”. Raw replicate values should be made available in supplementary materials.

Reviewer #3: Dear Author,

The author of an article entitled "Synergistic antifungal activity of antiretrovirals with amphotericin B against Aspergillus species". The concept of this manuscript is to highlight the potential of these drug combinations as promising treatment options for aspergillosis, leveraging already approved therapies. The current format will not be accepted. The author should carefully revise the reviewer comments and then resubmit.

Major revision:

1. The authors screened AmB among 618 compounds. However, since there are already several published articles demonstrating AmB’s antifungal activity, what is the novelty of this study?

2. Why did the authors choose Aspergillus species? In nature, several fungal species.

3. The authors have not explained how they selected the specific dose.

4. They should include an additional figure (e.g., as Figure 2) showing dose-dependent experiments to justify the chosen concentration. This information should also be clearly presented in the Materials and Methods section and discussed in the Results and Discussion.

5. The Materials and Methods section requires further elaboration to ensure clarity and reproducibility.

6. The authors should improve the quality of the figures before resubmission.

Best,

6. PLOS authors have the option to publish the peer review history of their article (what does this mean?). If published, this will include your full peer review and any attached files.

Reviewer #1: No

Reviewer #2: **Yes:**Sri Harshini Goli

Reviewer #3: **Yes:**Dr. Aabid Hussain

---

## [Author Response · Author response to Decision Letter 1]

5 Feb 2026

Letter with responses to queries and corrections has been attached in file titled "Response to Reviewers"

---

## [Decision Letter · Decision Letter 1]

19 Feb 2026

Synergistic antifungal activity of antiretrovirals with amphotericin B against Aspergillus species.

PONE-D-25-49174R1

Dear Dr. Seleem

We’re pleased to inform you that your manuscript has been judged scientifically suitable for publication and will be formally accepted for publication once it meets all outstanding technical requirements.

Kind regards,

Aijaz Ahmad, Ph.D.

Academic Editor

PLOS One

Additional Editor Comments (optional):

Reviewers' comments:

Reviewer's Responses to Questions

**Comments to the Author**

1. If the authors have adequately addressed your comments raised in a previous round of review and you feel that this manuscript is now acceptable for publication, you may indicate that here to bypass the “Comments to the Author” section, enter your conflict of interest statement in the “Confidential to Editor” section, and submit your "Accept" recommendation.

Reviewer #1: All comments have been addressed

Reviewer #2: All comments have been addressed

Reviewer #3: All comments have been addressed

2. Is the manuscript technically sound, and do the data support the conclusions?

Reviewer #1: Yes

Reviewer #2: Yes

Reviewer #3: Yes

3. Has the statistical analysis been performed appropriately and rigorously?

Reviewer #1: Yes

Reviewer #2: Yes

Reviewer #3: Yes

4. Have the authors made all data underlying the findings in their manuscript fully available?

Reviewer #1: Yes

Reviewer #2: Yes

Reviewer #3: Yes

5. Is the manuscript presented in an intelligible fashion and written in standard English?

Reviewer #1: Yes

Reviewer #2: Yes

Reviewer #3: Yes

6. Review Comments to the Author

Reviewer #1: I do not have any issue with revised draft and author addressed all concern raised in original submission. Draft looks ok for acceptance in its present revised from.

Reviewer #2: The Data Availability statement is acceptable in format; however, the provided S1 Data file contains primarily summary statistics (mean/SD) rather than the complete underlying raw datasets. For full reproducibility, the authors should provide replicate-level raw values for the OD530 growth kinetics and biofilm assays, and importantly the complete checkerboard datasets (MIC matrices / well-level growth endpoints) used to compute FICI values.

Reviewer #3: Dear Authors,

Thank you for revising and resubmitting your manuscript. It has now been accepted.

Congratulations.

7. PLOS authors have the option to publish the peer review history of their article (what does this mean?). If published, this will include your full peer review and any attached files.

Reviewer #1: **Yes:**RAVINDER KUMAR

Reviewer #2: **Yes:**Sri Harshini Goli

Reviewer #3: **Yes:**Aabid Hussain

---

## [Editor Report · Acceptance letter]

PONE-D-25-49174R1

PLOS One

Dear Dr. Seleem,

I'm pleased to inform you that your manuscript has been deemed suitable for publication in PLOS One. Congratulations! Your manuscript is now being handed over to our production team.

Kind regards,

on behalf of

Dr. Aijaz Ahmad

Academic Editor

PLOS One